

# An improved point cloud denoising method in adverse weather conditions based on PP-LiteSeg network

Wenzhen Zhang and Ming Ling

School of Electronic and Electrical Engineering, Shanghai University of Engineering Science, Shanghai, China

## ABSTRACT

Reliable point cloud data (PCD) generated by LiDAR are crucial to perceiving surroundings when autonomous driving systems are a concern. However, adverse weather conditions can impact the detection range of LiDAR, resulting in a significant amount of noisy data that substantially deteriorates the quality of PCD. Point cloud denoising algorithms used for challenging weather conditions suffer from poor accuracy and slow inferences. The manuscript proposes a Series Attention Fusion Denoised Network (SAFDN) based on a semantic segmentation model in real-time, called PP-LiteSeg. The proposed approach provides two key components to the model. The insufficient feature extraction issue in the general-purpose segmentation models is first addressed when dealing with objects with more noise, so the WeatherBlock module is introduced to replace the original layer used for feature extraction. Hence, this module employs dilated convolutions to enhance the receptive field and extract multi-scale features by combining various convolutional kernels. The Series Attention Fusion Module (SAFM) is presented as the second component of the model to tackle the problem of low segmentation accuracy in rainy and foggy weather conditions. The SAFM sequentially applies channel and spatial attention mechanisms to enhance the model's sensitivity to crucial features. Furthermore, weighted feature fusion is employed to enhance the model's efficiency in integrating low-level and high-level feature information configurations. Experimental evaluations were conducted on the publicly available DENSE dataset. The results demonstrate that the improved model achieved an 11.1% increase in denoising accuracy measured by MIOU and an inference speed of 205.06 FPS when compared to the PP-LiteSeg model. As a result, the noise recognition accuracy and denoising capability in real-time are enhanced.

## INTRODUCTION

High-precision LiDAR is often used to complement cameras, radar, ultrasonic sensors, and other devices to enhance the perception and understanding of surroundings in autonomous driving. Despite its high precision, noise is generated when functioning and sometimes fails to capture surroundings in real traffic scenes effectively. The point cloud noise (PCN) refers to the invalid points in the collected point cloud data (PCD) in LiDAR. The primary sources of PCN in LiDAR

Corresponding author
Ming Ling, 02180010@sues.edu.cn

can be attributed to several factors: (1) Adverse weather conditions, such as rain (*Kutila et al., 2018*), fog (*Bijelic, Gruber & Ritter, 2018*), and snow (*Charron, Phillips & Waslander, 2018*; *Heinzler et al., 2019*), can cause undesirable missing and scattered points due to laser reflection, refraction, and absorption. (2) Interference occurs between multiple LiDAR sensors, where a LiDAR sensor may receive pulses from other LiDAR sensors, which can lead to target loss or incorrect detection (*Kim et al., 2015*; *Hebel et al., 2018*). *Fersch et al. (2016)* researched LiDAR sensors with small apertures. The impact of rain is more severe with increased rainfall, and non-uniform heavy precipitation can cause the LiDAR point cloud to form clustered blocks, which may be misidentified as obstacles. *Zhang et al. (2023)* suggested that adverse weather conditions have long been a significant obstacle preventing autonomous vehicles from achieving Level 4 autonomy or higher, highlighting the severe impact caused by such conditions. Undoubtedly, noise or interference can increase the risk of traffic accidents, posing a threat to the safety of individuals on the road. These noises also affect other downstream perception algorithms, such as object detection (*Kilic et al., 2021*) and point cloud semantic segmentation (*Wang et al., 2018*).

Therefore, ensuring that autonomous vehicles can obtain reliable perception data for downstream sensing tasks in adverse weather conditions is of utmost importance. Real-time and efficient point cloud segmentation and denoising algorithms are needed. Among available ones, conventional sparse PCD approaches (*Charron, Phillips & Waslander, 2018*; *Rusu & Cousins, 2011*; *Kurup & Bos, 2021*; *Park, Park & Kim, 2020*) have demonstrated a certain level of effectiveness in denoising implementations. As deep learning algorithms advance, trained networks outperform conventional methods in terms of both segmentation quality and real-time performance. Common point cloud semantic segmentation methods include point-based approaches (*Seppänen, Ojala & Tammi, 2022*) and projection-based methods (*Heinzler et al., 2020*). When a spherical projection is used to transform 3D data into 2D images, a matured 2D real-time lightweight semantic segmentation network can also be applied to point cloud segmentation. Among these, the PP-LiteSeg network (*Peng et al., 2022*), known for its outstanding performance in image segmentation, employs novel modules like the Flexible and Lightweight Decoder (FLD) and Simple Pyramid Pooling Module (SPPM) to enhance segmentation accuracy while maintaining a competitive inference speed.

However, despite its exceptional inference speed, the segmentation performance of this model on datasets used for PCD is subpar. This could be attributed to the model's lack of adaptability to PCD's highly asymmetric aspect ratios and the more abstract and ambiguous noise characteristics when adverse weather conditions are observed. Thus, the article proposed an improved model based on PP-LiteSeg for PCD in adverse weather conditions. This model enhanced segmentation and PCD accuracy while balancing denoising accuracy and real-time performance. The experiments demonstrated that the proposed model outperformed WeatherNet (*Heinzler et al., 2020*) as well as leading semantic segmentation networks used for general purposes (*Wu et al., 2018*; *Milioto et al., 2019*) and exhibited strong generalization capabilities across various scenarios.

The contributions of the article can be summarized as follows:

1. An improved model is suggested based on PP-LiteSeg: A multi-scale initial feature extraction layer called WeatherBlock is proposed by addressing the extremely asymmetric aspect ratio of PCD and the more abstract nature of noise objects. This layer leverages dilated convolutions by stacking different convolution kernels to extract multi-scale features, enhancing the model's feature extraction capability. To further improve the fusion of low-level and high-level features and enhance denoising accuracy, the Series Attention Fusion Module (SAFM) is presented. The SAFM sequentially employs channel and spatial attention mechanisms to enhance the feature representation and achieve a better fusion of abstract features.

2. Comprehensive experiments are run: The publicly available DENSE dataset is tested extensively, and the proposed model is compared with other models used for filtering adverse weather conditions and leading generic semantic segmentation networks. Through these experiments, we evaluated the proposed model's denoising accuracy and real-time performance.

3. Ablation studies are conducted: The proposed enhancements were tested in ablation experiments, providing a more objective evaluation of the effectiveness of each improved component.

The objective is to advance the field of PCD in adverse weather conditions, providing a model that demonstrates improved accuracy while increasing real-time performance.

The rest of the article is presented as follows: the related work is presented in 'Related Work'. 'Materials and Methods' presents the fundamental material and methods and the proposed network. The experiments and their corresponding results are presented in 'Experimental Results and Discussion'. The research is concluded in 'Conclusions and Future Research'.

## RELATED WORK

### Conventional approaches

Conventional filtering approaches for PCD are typically based on spatial domains or statistical distributions. The Radius Outlier Removal (ROR) and Statistical Outlier Removal (SOR) algorithms proposed in *Rusu & Cousins (2011)* are well-known methods. The ROR algorithm identifies data points as outliers based on a radius value. If the distance between a data point and other points exceeds a radius threshold, the data point is considered an outlier and is removed. On the other hand, the SOR algorithm utilizes statistical characteristics of data points to identify outliers. It calculates the average distance and standard deviation between a data point and its neighbors and then uses a specific threshold to determine whether it is an outlier or not. However, due to the varying density of LiDAR PCD, such fixed radius thresholds may lead to removing genuine distant points, making them impractical for adverse weather denoising operations. *Charron, Phillips & Waslander (2018)* proposed the Enhanced Dynamic Radius Outlier Removal (DROR) algorithm, which employs a dynamically adjustable radius based on the data point distribution. This adaptive approach allows the algorithm to handle LiDAR PCD's density changes better, resulting in improved performance. *Kurup & Bos (2021)* introduced the Dynamic

Statistical Outlier Removal (DSOR) algorithm, which implements dynamic parameters to determine outlier points instead of conventional fixed thresholds. It calculates a dynamic threshold based on the mean and standard deviation of the data points, allowing the threshold to be adaptively adjusted based on the distribution of the data. *Park, Park & Kim (2020)* presented the Low-Intensity Outlier Removal (LIOR) algorithm, designed to remove outlier points caused by snow coverage in LiDAR PCD. This algorithm filters intensity values in the PCD based on an intensity threshold and employs the ROR filter to retain low-intensity points. Conventional filtering algorithms utilize spatial-domain information to remove noisy points. Although some algorithms can adaptively adjust certain thresholds, they still have limitations based on the distribution, and their denoising speed is generally slow.

## Trained approaches

Trained approaches, particularly those based on neural networks, have made significant strides in semantic segmentation applied to PCD. These approaches are tailored for general segmentation tasks, and they can be trained to segment highly abstract shapes, including noise patterns caused by rain and fog (*Seppänen, Ojala & Tammi, 2022*). Successful methods employ voxelization techniques, such as VoxNet, designed by *Maturana & Scherer (2015)*. These address the irregular structure of 3D PCD and achieve commendable results in the semantic segmentation tasks of PCD. However, their sparse nature can lead to subpar segmentation results when, for example, dealing with rain and fog noise. In addition, deep learning models operate entirely on PCD, such as PointNet (*Qi et al., 2017a*) and PointNet++ (*Qi et al., 2017b*). These models treat PCD as unordered sets of points and directly process their features, resulting in excellent performance in classification and segmentation tasks. However, using the original data points as input in these models can lead to high computational complexity and relatively lower real-time performance.

Furthermore, instead of point-based segmentation approaches, projection-based methods offer increased efficiency and improved real-time performance. The spherical projection method leverages the generation principle of LiDAR PCD (*Wu et al., 2018*), mapping each 3D point to a corresponding pixel on a 2D image. Employing this approach naturally transforms sparse and unstructured point clouds into dense and structured image data. *Wu et al. (2018)* introduced SqueezeSeg, which applies the lightweight SqueezeNet (*Iandola et al., 2016*) semantic segmentation network for segmentation operations and then reversely maps the segmentation results back to 3D space. Subsequent models like SqueezeSegV2 (*Wu et al., 2019*) and SqueezeSegV3 (*Xu et al., 2020*) incorporate the aggregated spatial-domain context and adaptive convolution modules, reducing interference caused by data distribution features and missing points. *Milioto et al. (2019)* proposed the fast RangeNet++, where designing a post-processing algorithm based on fast k-nearest neighbor search methods improves segmentation results. *Piewak et al. (2019)* introduced LiLaNet, which consists of continuous LiLaBlock units and delivers good segmentation performance but exhibits high computational complexity.

There are only a few studies on PCD networks used for adverse weather conditions. *Heinzler et al. (2020)* proposed WeatherNet, a projection-based method for noise

segmentation caused by rain and fog. When compared to leading general segmentation networks, their approach has fewer trainable parameters while achieving higher denoising performance and stability. However, the model has more continuous branching structures, leading to a slower inference speed. *Seppänen, Ojala & Tammi (2022)* introduced 4DenoiseNet, which utilizes a new k-nearest neighbor search convolution on continuous point clouds, incorporating temporal-dimension information to improve denoising performance for snow. However, its limitation lies in the requirement for PCDs with temporal continuity.

Semantic segmentation networks have been extensively validated and shown to be versatile. Therefore, it is feasible to train real-time lightweight semantic segmentation models with superior performance to segment images to determine noise caused by adverse weather conditions. Among them, the PP-LiteSeg model (*Peng et al., 2022*) incorporates novel modules such as the Flexible and Lightweight Decoder (FLD) and Simple Pyramid Pooling Module (SPPM) to enhance image segmentation accuracy, and it exhibits advantages in terms of operating speed when compared to real-time semantic segmentation models. However, its direct application to PCD tasks yields subpar results. To better adapt to PCD tasks under adverse weather conditions, the article introduces two improvement strategies to the initial feature extraction layer and feature fusion module built upon PP-LiteSeg. These strategies effectively enhance the segmentation accuracy of noisy points, striking a balance between real-time performance and denoising precision.

## MATERIALS AND METHODS

### Dataset

The DENSE dataset is a collaborative effort between Mercedes-Benz, Algolux, Ulm University, and Princeton University (*Gruber et al., 2019*). The dataset specifically focuses on adverse weather conditions and is the first large-scale multi-modal dataset. The research primarily utilized a subset of the DENSE dataset known as "Point Cloud Denoising" (*Heinzler et al., 2020*). This subset is specially curated for the challenging task of denoising LiDAR point clouds in adverse weather conditions. As depicted in Fig. 1, the dataset includes four real-world scenarios: (Fig. 1A) pedestrian crossing, (Fig. 1B) construction area, (Fig. 1C) highway, and (Fig. 1D) pedestrian zone. Each scenario is represented under various visibility levels in different weather conditions (day/night) with clear weather, rain, and fog. The PCD dataset consists of approximately 72,800 PCD samples with a resolution of $2\times 32\times 400$. The dataset includes four semantic classes for labeling: "valid point" (representing points unaffected by adverse weather), "fog" (indicating points affected by fog clutter), "rain" (representing points affected by rain clutter), and "invalid point" (points labeled as invalid for other reasons). To enhance the generalization capability and robustness of the model, we applied data augmentation techniques, including horizontal flipping.

### Point cloud denoising method based on improved PP-LiteSeg

To better adapt the model to the PCD task, an improved model based on PP-LiteSeg is proposed. Firstly, we enhanced the model's competency to fuse high-level and low-level

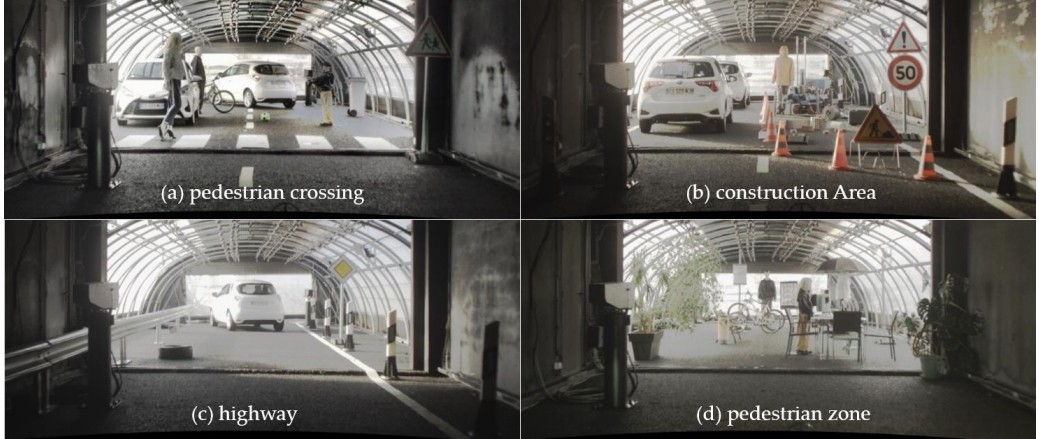

**Figure 1  Static setups in the climate chamber representing four complex and realistic traffic scenarios.** (A) pedestrian crossing, (B) construction area, (C) highway and (D) pedestrian zone. Image source: DENSE dataset, *Gruber et al. (2019)*.

features by employing spatial and channel attention mechanisms. Secondly, we enhanced the initial feature extraction layer to improve the model's feature extraction ability, enabling it to capture abstract features of rain and fog-related noisy data and address point clouds' extremely asymmetric aspect ratio. The following sections will provide detailed information on these improvements.

### *Overall architecture of the PP-LiteSeg model*

PP-LiteSeg (*Peng et al., 2022*) is a versatile, efficient, and real-time semantic segmentation model with an overall architecture, as illustrated in Fig. 2. It follows a typical encoder–decoder structure, composed of three main components: an encoder based on the efficient STDCSeg backbone feature extraction network (*Fan et al., 2021*), a Simple Pyramid Pooling Module (SPPM), and a lightweight decoder with the Unified Attention Fusion Module (UAFM). We employed the images generated from the spherical projection of the PCD as inputs to the model, encompassing information from both the distance and intensity channels similar to the data creation process outlined in *Wu et al. (2018)*. The encoder employs the STDCSeg backbone feature extraction network to output abstract high-level feature maps at a lower computational cost. The SPPM aggregates semantic information from multiple scales to extract and fuse context information. It utilizes addition to replace the original cascaded calculations, simplifying the computational complexity. Then, It feeds the fused composite feature maps into the decoder. The decoder incorporates two SAFM (Series Attention Fusion Module) blocks and a segmentation head. It progressively fuses multi-level features containing low-level and high-level feature information. It employs upsampling operations to restore the feature maps to the original resolution, resulting in

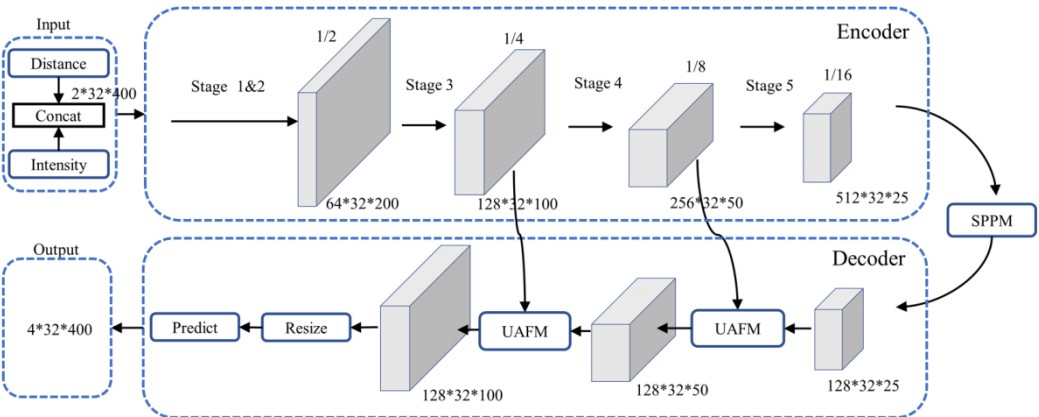

**Figure 2** Architecture of PP-LiteSeg.

the output of the predicted segmentation outcomes. The output comprises four channels, each corresponding to the segmentation results of the four classes in the dataset.

The encoder in Fig. 2 consists of multiple STDC modules organized into different stages. STDC (*Fan et al., 2021*), while extracting low-level detailed features, employs a lightweight design by gradually reducing the number of feature channels as the network deepens. Moreover, the output of the STDC module comprises multiple feature maps, enabling it to extract multi-scale features with fewer parameters. However, the initial feature extraction layers in Stages 1 and 2 consist of two 3 × 3 convolutional kernels, which may exhibit limited feature extraction capabilities when dealing with input tensors characterized by low resolution and highly asymmetric aspect ratios. The innovative Unified Attention Fusion Module (UAFM) in a PP-LiteSeg is designed to merge the details from lower layers with the semantic information of higher layers, as depicted in Fig. 3A. The UAFM initially upsamples the feature maps (Fup) from the deep modules in the decoder to match the size of the corresponding feature maps in the encoder (Flow). Hence, attention modules take Fup and Flow as inputs, generating attention weights ($\alpha$). These weights are then used to fuse Fup and Flow through element-wise multiplication (Mul operation). Finally, the weighted features are element-wise added, resulting in the output of the fused features. The above process can be described in Eq. (1):

$$F_{up} = Upsample\left(F_{high}\right)$$
$$\alpha = Attention\left(F_{up}, F_{high}\right) \tag{1}$$
$$F_{out} = F_{up} \cdot \alpha + F_{low} \cdot (1 - \alpha)$$

The attention module, as illustrated in Fig. 3B, can be considered a plugin. The spatial attention module initially computes the average and maximum values along the channel dimension to generate four features. Then, a concatenate operation is performed, followed by a convolutional operation, and the sigmoid function is applied to obtain the attention weight. The channel attention module utilizes average pooling and max pooling operations

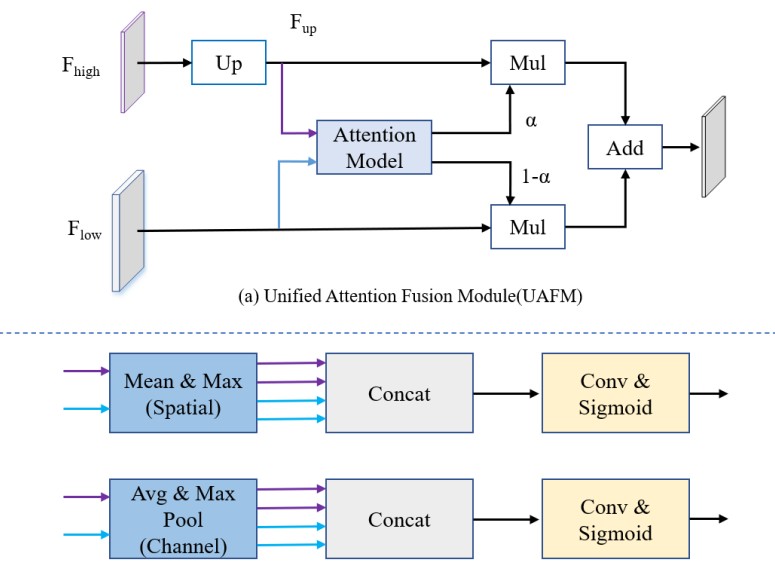

(a) Unified Attention Fusion Module(UAFM)

(b) Spatial Attention Module and Channel Attention Module

**Figure 3** **(A) The framework of the Unified Attention Fusion Module (UAFM). (B) Spatial attention module and channel attention module for plugins.** The UAFM applies plugins to generate weights α for fusing the low-level and high-level features.

to compress the spatial dimension of the input features. Similarly, a convolutional operation is performed after the concatenate operation, and the Sigmoid function is applied to obtain the attention weight.

### The improved PP-LiteSeg model

Although the PP-LiteSeg performs better in general segmentation tasks, its direct application to PCD under adverse weather conditions yields suboptimal results. It tends to misinterpret or introduce confusion when dealing with noise caused by rain and fog conditions. To address these issues, the article proposes an enhanced model based on PP-LiteSeg, which is referred to as the Series Attention Fusion Denoised Network (SAFDN). This architecture, as depicted in Fig. 4, incorporates the Series Attention Fusion Module (SAFM) and the WeatherBlock module.

We enhanced the structure of the UAFM, inspired by the work of *Woo et al. (2018)*, as illustrated in Fig. 5. Specifically, we sequentially applied attention mechanisms along the channel and spatial dimensions. This strategy guides the neural network to focus more effectively on semantic categories relevant to the segmentation task, such as rain and fog-related noisy data, enhancing responsiveness to crucial features. The improved module enriches the fusion representation of low-level and high-level features without significantly lowering the network's operational speed, thus enhancing denoising performance. The channel attention module in the diagram outputs weight $\alpha_c$, while the spatial attention module outputs weight $\alpha_s$. The computational flow is akin to the original UAFM structure,

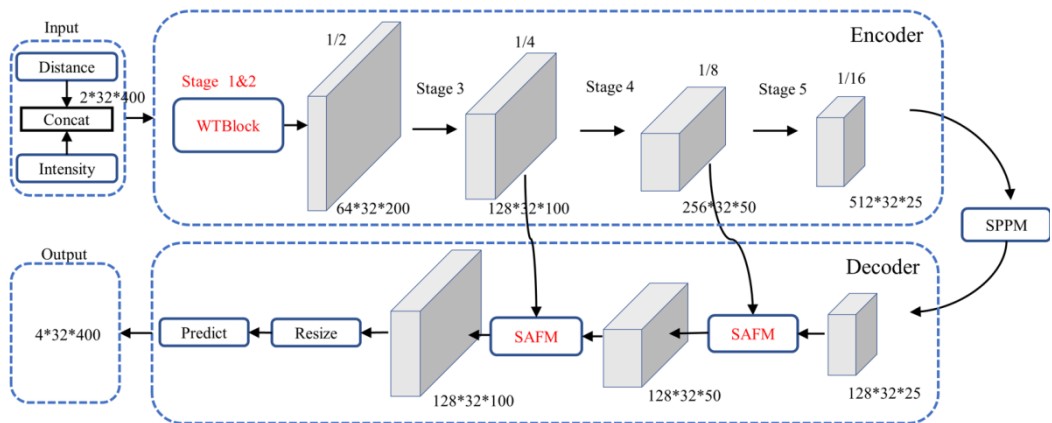

**Figure 4** Architecture of the improved PP-LiteSeg.

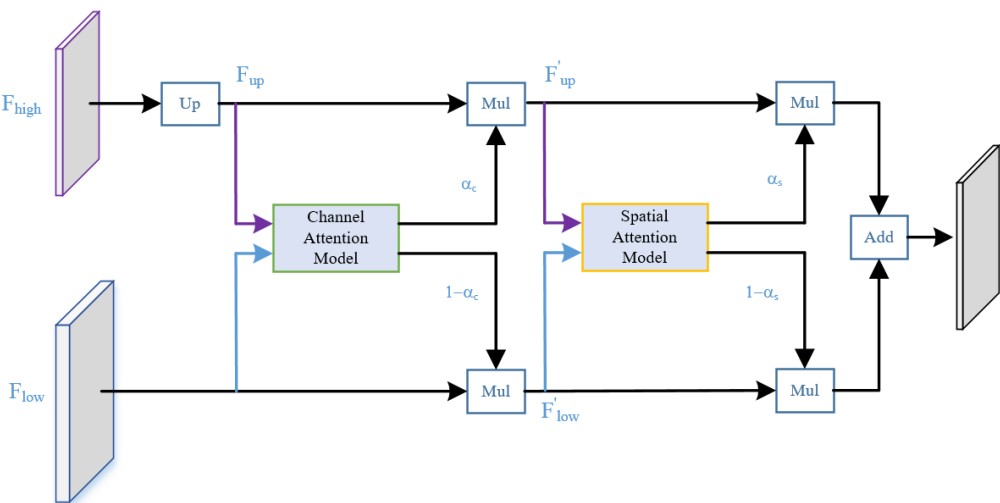

**Figure 5** Structure of the series attention fusion module (SAFM).

and this process can be expressed using Eq. (2).

$$
\begin{aligned}
F_{up} &= Upsample\left(F_{high}\right) \\
\alpha_c &= Attention_c\left(F_{up}, F_{low}\right) \\
F'_{up} &= F_{up} \cdot \alpha_c, \quad F'_{low} = F_{low} \cdot (1 - \alpha_c) \\
\alpha_s &= Attention_s\left(F'_{up}, F'_{low}\right) \\
F_{out} &= F'_{up} \cdot \alpha_s + F'_{low} \cdot (1 - \alpha_s)
\end{aligned}
\tag{2}
$$

where the attention function applies the two modules in Fig. 3B.

We designed an enhanced multi-scale initial feature extraction layer called WeatherBlock, inspired by both LiLaBlock (*Piewak et al., 2019*) and WeatherNet (*Heinzler et al., 2020*), to address the challenges posed by low-resolution LiDAR point cloud images and input

**Peer**J Computer Science

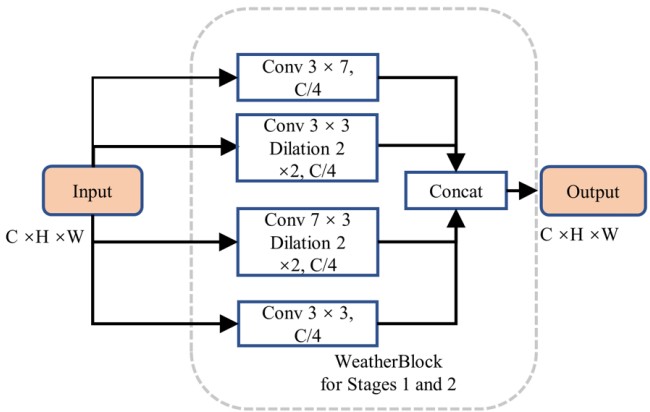

**Figure 6** Structure of WeatherBlock in the initial feature extraction layer.

tensors with imbalanced aspect ratios as shown in Fig. 6 in the encoder. To handle various aspect ratios of relevant objects more effectively, this module applies parallel convolutions with kernel sizes of 7 × 3, 3 × 7, and 3 × 3, respectively, thus increasing the network's adaptability to different scales. Due to the different receptive fields in these branches, multi-scale information is incorporated. In the Conv 7 × 3 convolution module, we additionally introduced dilated convolutions. This augmentation provides more spatial proximity information without significantly increasing the number of parameters. To balance computational complexity and feature extraction capability, we reduced the channel numbers and performed downsampling on the intermediate feature maps, reducing the number of the parameters and computational burden of this layer. Subsequently, we employed the concatenate operation to fuse information from the four intermediate feature maps, restoring the feature map to the original channel size. Both Stages 1 and 2 share the same structure; however, Stage 2 outputs a downsampled feature map, reducing the width of the feature map to half of the input. Additionally, each convolution is followed by a Rectified Linear Unit (ReLU) layer to represent non-linear relations. Finally, we incorporated Dropout layers with a dropout rate of 0.5 in each layer to prevent overfitting, enhancing the model's generalization capability.

## Performance evaluation

We employed Intersection over Union (IOU) and Mean Intersection over Union (MIOU) as performance evaluation metrics for the trained model, focusing solely on the categories of valid points and noise points, respectively. These metrics are defined by Eqs. (3) and (4).

$$IOU = \frac{TP}{TP + FP + FN} \tag{3}$$

$$MIOU = \frac{1}{k+1} \sum_{i=0}^{k} IOU_k \tag{4}$$

where TP represents true positives, which are correctly classified as valid or noise points; FP represents false positives, incorrectly classified as valid or noise points; and FN represents false negatives, indicating points that should have been classified as valid or noise but actually were not.

In addition, the model's parameter size, floating-point operation numbers (FLOPs or GFLOPs), and inference speed (FPS) were implemented to assess the size and real-time performance of the model.

## Experimental preparation

The research maintained identical configurations, parameter settings, and datasets used for all models to ensure the comparability of the denoising results across different models. To mitigate the impact of inter-sample correlations within the same scene, we selected data from scenarios Fig. 1C and Fig. 1D for training, data from scenario (Fig. 1B) for validation, and data from scenario (Fig. 1A) for testing, respectively. The data distribution for training, validation, and testing was approximately set to 60%, 15%, and 25%, respectively. Additionally, data augmentation was run by employing horizontal flipping. All experiments were conducted on an Ubuntu 22.04 system with an Intel(R) Core CPU i5-9400F @ 2.90 GHz and an NVIDIA GeForce GTX 1660 SUPER.

The experiments utilized the standard cross-entropy loss function, employing the Adam optimizer for parameter updates. A batch size of 16 was set, and the initial learning rate was chosen $1 \times e^{-4}$, with a decay factor of 0.90 after each epoch. The recommended default values for the Adam optimizer were specifically, $\beta_1 = 0.9$, $\beta_2 = 0.999$, and $\varepsilon = 10^{-8}$, respectively employed (*Kingma & Ba, 2014*).

## EXPERIMENTAL RESULTS AND DISCUSSION

To evaluate the effectiveness of the proposed method, comprehensive experiments were conducted on the publicly available DENSE dataset. These experiments included comparative analyses with WeatherNet and advanced general-purpose semantic segmentation networks used for adverse weather denoising. Additionally, ablation experiments were performed to assess the individual contributions of the proposed modules.

## Quantitative results

We employed the same training methodology to compare the improved model with leading semantic segmentation models. The comparison included popular models such as WeatherNet, SqueezeSeg, RangeNet21, LiLaNet, and the baseline of PP-LiteSeg, where the spatial attention module is utilized in the UAFM of PP-LiteSeg. The experimental results are summarized in Table 1.

Table 1 presents the performance comparison of the proposed SAFDN model with other models in the MIOU (Mean Intersection over Union), showing that SAFDN achieved the highest MIOU score, demonstrating superior denoising effectiveness. Specifically, when compared to the mentioned models, the MIOU score of SAFDN outperformed that of the other models by 3.0%, 20.2%, 11.1%, 7.8%, and 4.3%, respectively.

**Table 1   Performance comparison of the different models.**

| Model | GFLOPs | Parameters/M | Inference speed/FPS | IOU/% | | | MIOU/% |
|---|---|---|---|---|---|---|---|
| | | | | Valid | Fog | Rain | |
| WeatherNet (*Heinzler et al., 2020*) | 18.42 | 1.53 | 59.69 | 93.2 | 83.6 | 92.3 | 89.7 |
| SqueezeSeg (*Wu et al., 2018*) | 1.25 | 0.90 | 109.46 | 93.5 | 54.4 | 69.50 | 72.5 |
| RangeNet21 (*Milioto et al., 2019*) | 59.30 | 38.28 | 56.45 | 86.7 | 80.9 | 87.0 | 84.9 |
| LiLaNet (*Piewak et al., 2019*) | 71.88 | 7.84 | 20.87 | 91.6 | 84.9 | 88.6 | 88.4 |
| PP-LiteSeg (*Peng et al., 2022*) | 4.11 | 3.53 | 229.25 | 92.1 | 72.9 | 79.6 | 81.6 |
| SAFDN (ours) | 8.14 | 3.60 | 205.06 | 96.4 | 88.0 | 93.7 | 92.7 |

While the proposed model exhibited higher parameter numbers than the adverse weather denoising model WeatherNet, our encoder, which utilizes the lightweight STDC backbone network, incorporates the WeatherBlock module to extract features without additional multi-branch structures. As a result, the floating-point operation number was only 44.2% that of WeatherNet. SqueezeSeg boasted the lowest GFLOPs and parameter numbers, showcasing remarkable lightweight characteristics. However, its MIOU score was the lowest at 72.5%. The complexity of the model is compressed and the effectiveness of semantic segmentation is diminished due to the model's decoder module.

The proposed model consistently outperformed PP-LiteSeg, with a significant 15.1% IOU improvement for the fog category. This can be attributed to the effective feature extraction competency of the WeatherBlock module in the initial layers and the capability of the SAFM to fuse high-level and low-level features concurrently. These results validate the efficacy of the WeatherBlock module in feature extraction and the effectiveness of SAFM in high-level and low-level feature fusion operations.

Although both RangeNet21 and LiLaNet achieved high MIOU scores, the former had higher parameter numbers and GFLOPs, and the latter exhibited the slowest inference speed, making it less suitable for real-time PCD in the preprocessing stage. Note that the proposed model's inference speed was lower than that of PP-LiteSeg due to the more complex initial feature extraction module and increased computational costs. However, it still outperformed the other models, maintaining a real-time performance of over 200 FPS.

Table 1 indicates that the various models performed reasonably well in distinguishing valid points, but most struggled to segment rain and fog-related noisy data effectively. Figure 7 depicts that the confusion matrices reveal the degree of confusion that some well-performing models exhibit worse performance when dealing with rain and fog-related noisy data. The fact that fog and rain consist of water droplets and differ mainly in density and size decreases the performance of neural networks. However, when compared to PP-LiteSeg and WeatherNet, the proposed model reduced the misclassification rate of rain-related noisy data over fog-related noisy data by 1.12 and 1.97, respectively. Overall, the proposed model exhibited smaller confusion levels than the other models and achieved the best performance regarding accuracy.

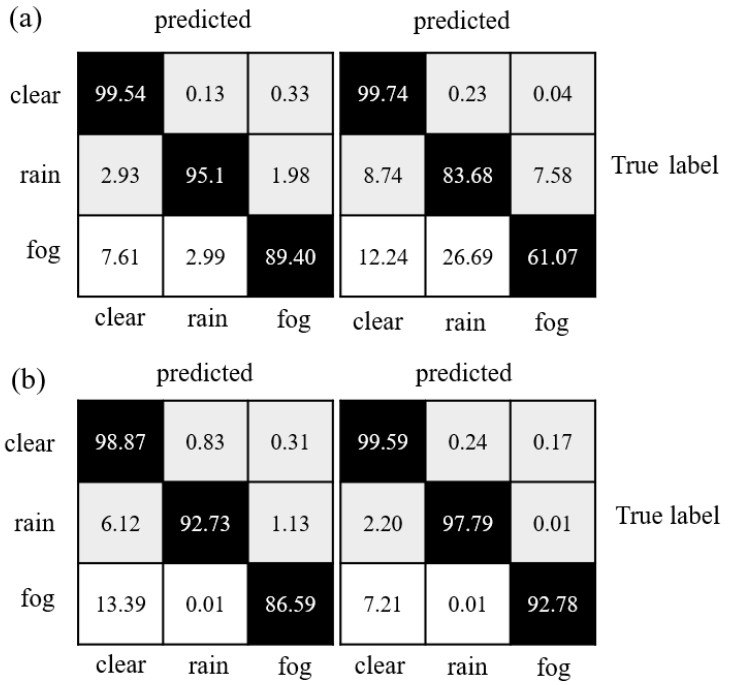

**Figure 7** Confusion matrix comparison of model segmentation results: (A) WeatherNet and Squeeze-Seg; (B) PP-LiteSeg and SAFDN (ours).

**Table 2 Ablation experiment results.**

| Model | WeatherBlock | SAFM | FLOPs/M | Parameters/M | IOU/% | | | MIOU/% | Inference speed/FPS |
|---|---|---|---|---|---|---|---|---|---|
| | | | | | Valid | Fog | Rain | | |
| Baseline | – | – | 4114.47 | 3.53 | 92.1 | 72.9 | 79.6 | 81.6 | 229.23 |
| SAFDN | ✓ | – | 8143.07 | 3.55 | 95.6 | 84.5 | 89.2 | 89.8 | 208.57 |
| SAFDN | – | ✓ | 4114.52 | 3.58 | 93.7 | 75.7 | 81.6 | 83.6 | 222.56 |
| SAFDN | ✓ | ✓ | 8143.12 | 3.60 | 96.4 | 88.0 | 93.7 | 92.7 | 205.06 |

## Ablation experiment and result

To objectively assess the effectiveness of the proposed method and evaluate the impact of each improved module on the model performance, we conducted ablation experiments on the improved modules under the same conditions. The experimental results are shown in Table 2, where the baseline model represents the original PP-LiteSeg without any introduced improvements, and the UFAM in the baseline model employs the spatial attention module. The "✓" sign indicates the inclusion of the respective improvement strategy.

(1) Influence of WeatherBlock on the model performance

As shown in Table 2, the first row represents the denoising by using the baseline model, with IOU values of 92.1% for valid points, 72.9% for fog-related points, and 79.6% for rain-related points, respectively. When compared to the baseline model, the IOU for

each category increased by 3.5%, 11.6%, and 9.6%, respectively, with the most significant improvement in the detection performance of fog-related noisy data after the WeatherBlock module was incorporated. The MIOU increased by 8.2%, indicating that, the WeatherBlock module contributed to expanding the network's receptive field in the initial layers when compared to the original $3 \times 3$ convolution operation in the initial feature extraction layer of the baseline model. When rain and fog feature information at different scales are extracted better, the model's competency is effectively enhanced to identify rain and fog-related noise points. On the other hand, the fact that the WeatherBlock module includes four parallel convolutional operations, the increase in recognition accuracy is accompanied by a rise in the model's FLOPs, leading to a greater demand for computational resources. Although the inference speed decreases by 9.0%, the increase in computable parameters becomes minimal, with only 0.02M. Overall, the WeatherBlock module, while increasing the FLOPs and sacrificing detection speed, significantly enhances the denoising accuracy.

(2) Influence of the SAFM on model performance

Since the SAFM is incorporated, the model applies spatial attention and channel attention mechanisms sequentially during feature fusion operation. This effectively emphasizes relevant feature information for noise removal, enriches the fusion of low-level and high-level feature maps, and enhances the feature representation of rain and fog-related noise objects. As shown in the third row of Table 2, this enhancement improved recognition accuracy for each object category when compared to the baseline model. Specifically, the IOU for the valid, fog and rain categories increased by 1.6%, 2.8%, and 2.0%, respectively, resulting in a 2.0% improvement in MIOU. While there was a slight increase in the FLOPs and parameter numbers, the inference speed decreased by approximately 2.9%. This indicates that the real-time performance of the model remained high, even with the addition of the SAFM, which effectively enhanced both the segmentation and denoising performance without significantly increasing the model complexity or computational requirements.

By introducing both the WeatherBlock module and SAFM, the model built upon a more comprehensive initial feature extraction layer allowed for a more effective emphasis on the fusion of both low-level and high-level feature information. This resulted in a further improvement in accuracy for the recognition of various target objects. Specifically, the IOU for the valid, fog and rain categories increased by 4.3%, 15.1%, and 14.1%, respectively, resulting in an 11.1% improvement in MIOU. Notably, the most significant enhancement in detection accuracy was observed for fog-related noise points. On the other hand, due to the increased complexity of the network, the model's inference speed decreased by 10.5%, yet it still maintained a real-time performance of over 200 FPS. Overall, the experimental results demonstrated that although there was a trade-off in terms of an increase in the FLOPs and a minor decrease in detection speed, the proposed model successfully achieved a balance between real-time performance and denoising accuracy.

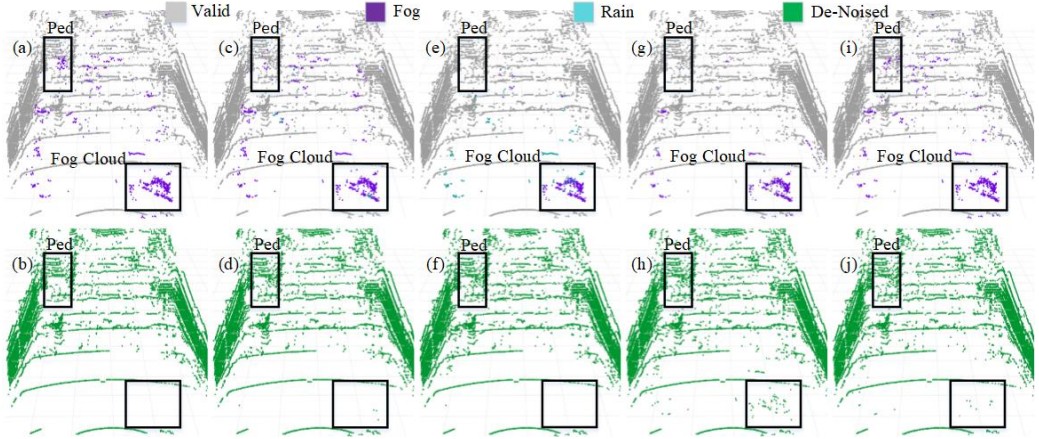

**Figure 8** Segmentation and denoising results for each model in the fog environment of scene.

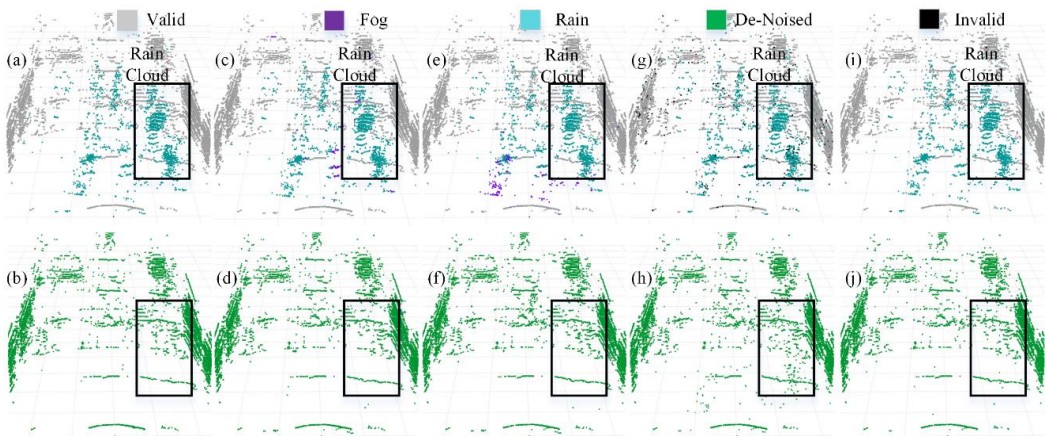

**Figure 9** Segmentation and denoising results for each model in the rainfall environment of scene.

## Qualitative results and discussion
### Qualitative results on chamber data

We selected four representative models to visualize a more intuitive comparison of the denoising performance in adverse weather conditions. The PCD used for visualization was taken from the scences in Fig. 1A and Fig. 1B. The original PCD and the qualitative visualization of the denoising results are depicted in Figs. 8 and 9, respectively. Black boxes highlight pedestrians and distinct fog clutter in Fig. 8 and rain clutter in Fig. 9, respectively.

The four models showcased in Figs. 8 and 9 demonstrate the capability to remove noise in rainy and foggy scenes, but the denoised results vary. In Fig. 8, the proposed model exhibits a more accurate identification of fog-related noise near distant pedestrians, and it shows a significant improvement over PP-LiteSeg in the recognition accuracy of nearby fog-related noise. WeatherNet also performs well in denoising nearby areas but tends to

exhibit noticeable confusion between rain and fog labels. Note that SqueezeSeg achieves the best denoising results for nearby regions but struggles to recognize noise in the middle to distant areas, especially near pedestrians. In comparison, the proposed model's denoising performance in nearby areas is less impressive, potentially due to the loss of some essential information during the upsampling process in the decoder, which restores the feature map to the original resolution. In Fig. 9, the proposed model can more effectively identify and remove noise caused by rainfall when compared to PP-LiteSeg within the black boxes.

Additionally, PP-LiteSeg misclassifies some valid points and rain-related noise points as invalid points. While WeatherNet and SqueezeSeg exhibit good denoising performance, they similarly confuse rain and fog labels. Qualitatively, the proposed model demonstrates better denoising performance for noise caused by rain and fog conditions.

### Qualitative results on dynamic road data

Additionally, we conducted qualitative testing on LiDAR PCD collected from real dynamic roads with different scenarios under adverse weather conditions. The selected PCD under rainy weather conditions were sourced from publicly available datasets, namely nuScense (*Caesar et al., 2020*) and Radiate (*Sheeny et al., 2021*). Although nuScenes includes a noise label, these chose PCD lacking specific Ground Truth labels for noise points regarding rain and fog categories. As employed in *Heinzler et al. (2020)* for static scenes, the automatic labeling method does not apply to the dynamic road PCD mentioned above. The manual labeling of such data becomes exceedingly challenging. Therefore, we conducted qualitative assessments. Figure 10 displays the segmentation results for the baseline model PP-LiteSeg (top row) and the improved model SAFDN (bottom row) under rainy conditions. By contrasting Fig. 10A and Fig. 10B, the proposed model effectively detects rain-related noise and reduces false negatives in object detection, such as pedestrians on the roadside. Furthermore, by comparing Fig. 10B and Fig. 10D, the proposed model yields improved segmentation results to detect the number of raindrops. Based on visual analysis of the noise segmentation results, the proposed model performs better in real-world scenarios with rain, showcasing its ability to generalize across different scenarios.

## CONCLUSIONS AND FUTURE RESEARCH

The fast and accurate removal of severe noise when adverse weather conditions exist is crucial for the safety of autonomous driving systems. In this article, we propose a real-time PCD model based on improvements to PP-LiteSeg. By enhancing the initial feature extraction layer and feature fusion module, the model significantly improves the segmentation and denoising accuracy without severely reducing the inference speed. The experimental results demonstrate that the proposed model accurately segments and removes noise points caused by rain and fog conditions and generalizes well to various road scenarios. The IOU for the valid point, fog, and rain categories were improved by 4.3%, 15.1%, and 14.1%, respectively, with an overall MIOU improvement of 11.1%. Furthermore, the inference speed reached 205.06 FPS, balancing denoising accuracy and

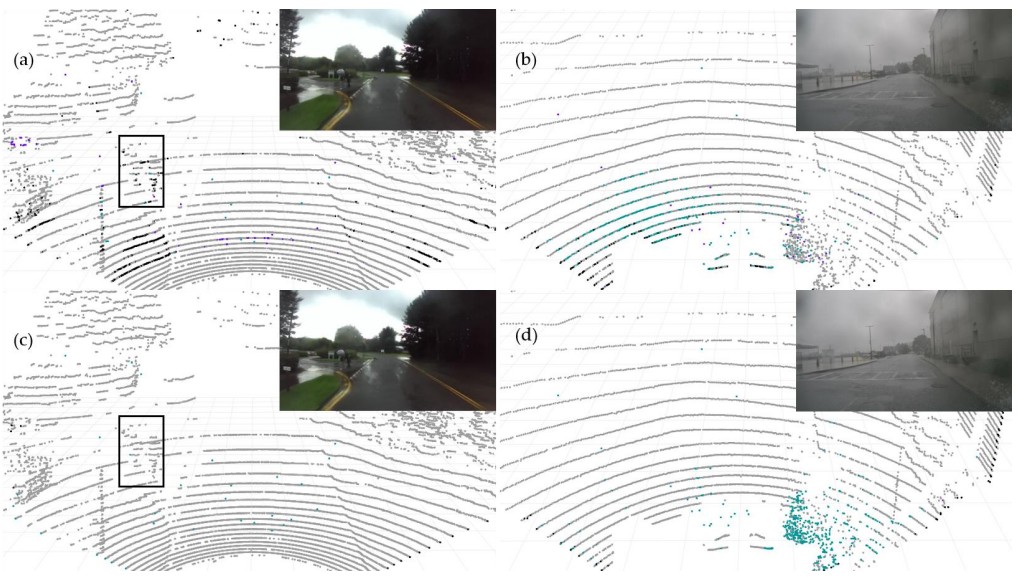

**Figure 10** **The baseline model PP-LiteSeg (A and B) and the proposed model SAFDN (C and D) compared in terms of noise segmentation in two individual sample point cloud data from real rainfall.** Image source: DENSE dataset, (*Gruber et al., 2019*).

real-time performance concurrently. This model suits segmentation and noise removal tasks under adverse weather conditions.

In future work, to enhance the adaptability of the denoising model to other adverse weather conditions, we will consider expanding the scope of the research to include snowy weather conditions, considering their unique characteristics and challenges.

### Funding
This study was supported by the Technical Standard Project Foundation of Shanghai (No. 21DZ2204300). The funders had no role in study design, data collection and analysis, decision to publish, or preparation of the manuscript.

### Grant Disclosures
The following grant information was disclosed by the authors:
The Technical Standard Project Foundation of Shanghai: 21DZ2204300.

### Competing Interests
The authors declare there are no competing interests.

### Author Contributions
- Wenzhen Zhang conceived and designed the experiments, performed the experiments, analyzed the data, performed the computation work, prepared figures and/or tables, authored or reviewed drafts of the article, and approved the final draft.

- Ming Ling conceived and designed the experiments, performed the experiments, analyzed the data, performed the computation work, prepared figures and/or tables, authored or reviewed drafts of the article, and approved the final draft.

## Data Availability

The data and the code are available in the Supplementary Files. The data was a subset of the DENSE dataset (A Point Cloud Denoising Method in Adverse Weather): Available at https://github.com/codekztwo/SAFDN/tree/main. The DENSE dataset is available at Available at https://www.uni-ulm.de/in/ui-drive-u/projekte/dense-datasets/.

## Supplemental Information

Supplemental information for this article can be found online at http://dx.doi.org/10.7717/peerj-cs.1832#supplemental-information.

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
