# Peer review of "An improved point cloud denoising method in adverse weather conditions based on PP-LiteSeg network"

_PeerJ Computer Science, doi:10.7717/peerj-cs.1832_

## Round 0.1 · original submission · Major Revisions

Dear authors

Your manuscript has been reviewed by experts in the field and you will see that they have concerns about the current quality of the paper.

Therefore to consider it further, please carefully revise the paper in light of the comments provided and resubmit.

**Language Note:** The review process has identified that the English language must be improved. PeerJ can provide language editing services - please contact us at copyediting@peerj.com for pricing (be sure to provide your manuscript number and title). Alternatively, you should make your own arrangements to improve the language quality and provide details in your response letter. – PeerJ Staff

·

Basic reporting

a. The title of the article should be checked and reflect the contribution of the research
b. All titles should be checked and fixed where necessary since some of them do not reflect what will be presented.
c. The title has this abbreviation, “Based on PP-LiteSeg”. Generally, abbreviations are not used in the title of the article. It should be given fully.
d. This sentence is extracted from the article: “The main work of this paper can be summarized as follows:” The contribution of the article should be better stated.

Experimental design

e. “Adverse weather conditions” are a very generic term. Instead of using this, specific weather conditions should be underlined.
f. A section should be allocated to the proposed method.
g. All terms in all equations should be explained just below each equation since what these equations say is incomprehensible.
h. Why did the authors select the dropout rate of 0.5_ how did they decide? Is there a piece of evidence from the data or empirically picked?

Validity of the findings

i. What are the percentages allocated to training and test data sets? Please provide.
j. Why did the authors pick 5 other models to compare with the proposed model? Are they all in the literature? Please have more discussion.

Additional comments

k. The speed of the proposed method is not good. How did the author explain this deficiency? Please provide more information.
l. Since the speed of the proposed method is relatively worse than others except for PP-Litesg. Could the proposed method be applied to real-life implementations? Please discuss it.
m. What are the disadvantages of the proposed method? Please discuss it.

Reviewer 2 ·

Basic reporting

The authors contributed to the available literature. However, the detected issues need to be fixed carefully to improve the presentation, technical competence, and readability of the article. A major revision is required.

1. All equations should be separately numbered.
2. All terminology and notation should be clearly expressed just below each equation.
3. Authors use several abbreviations. They need to be presented fully before the abbreviations are used in the text.
4. Why did the authors pick the IOU metric? Please discuss more.
5. Instead of using “ours”, the proposed method should be used.
6. proofreading is a must
7. The total number of words in the abstract is more than a standard abstract. The abstract should be shortened. The content of the abstract should include the research motivation, the proposed method, the data, and key results. The current abstract has redundancies that need to be removed.
The abstract should be rewritten and reorganized.
8. At the end of the introduction section, how the rest of the article is outlined should be written.
9. Some paragraphs are very long. They need to be shortened by preserving the content information.
10. How are outliers generated in Lidar applications? Please discuss.
11. The title of subsection 2.2 should be changed.
12. For what purposes did the authors add the attention fusion model to the algorithm? Please discuss it.
13. What advantages are brought to the model when average and max pooling are used together? Please discuss it
14. Why did the authors choose the sigmoid function to attain attention weights? Please discuss it.
15. The number of more up-to-date references should be added to and discussed.

Experimental design

.

Validity of the findings

.

Reviewer 3 ·

Basic reporting

The paper presents the Series Attention Fusion Denoised Network (SAFDN), an improved version of the PP-LiteSeg model, designed to enhance the quality of LiDAR point cloud data in autonomous driving systems. The model incorporates the WeatherBlock module and the Series Attention Fusion Module (SAFM) for enhanced feature extraction and increased segmentation accuracy under conditions of rain and fog noise. Tested on the DENSE dataset, the model demonstrated an 11.1% improvement in denoising accuracy and an inference speed of 205.06 FPS, effectively balancing real-time performance with denoising precision. This improvement is particularly significant for the performance of LiDAR in autonomous driving under adverse weather conditions.

Experimental design

To further enhance the quality of this paper, the following suggestions are proposed:
1. Diversification of Datasets: Although the DENSE dataset focuses on adverse weather conditions, incorporating a more diverse range of datasets could improve the model's generalization ability across various environments and conditions.
2. Optimization of the Feature Extraction Layer: The current feature extraction layer may have limitations when dealing with highly asymmetric input tensors. Exploring more advanced convolutional structures, such as deformable convolutions, could be more effective in capturing complex features.
3. Innovative Application of Deep Learning Techniques: Consider the adoption of the latest deep learning technologies, such as Transformers or Graph Neural Networks (GNNs), which have shown superiority in feature extraction and pattern recognition.
4. Balancing Real-Time Performance and Accuracy: Despite the model achieving high computational speed, there is room for improvement in balancing real-time performance and accuracy. Network pruning or quantization techniques could be used to further optimize the network for a better balance between speed and accuracy.

Validity of the findings

Research could consider the following directions to improve the point cloud denoising model:
1. Dataset Diversification: Incorporate a wider variety of datasets, such as road conditions from different countries and regions, to enhance the model's global generalization capability.
2. Algorithm Optimization: Explore more efficient algorithms, like lightweight networks in deep learning, to further increase inference speed and reduce computational resource consumption.
3. Model Robustness: Investigate the model's performance under extreme weather conditions (such as heavy snow or strong winds) to ensure its stability and reliability under all conditions.
4. Real-time Feedback Mechanism: Develop a real-time feedback mechanism to quickly adjust parameters or strategies when the model encounters predictive difficulties.
5. Cross-domain Applications: Explore the potential applications of the model in other domains, such as drone navigation or robot vision, to expand its range of applications.

---

## Round 0.2 · accepted · Accept

Thank you for your revised version submission, the reviewers are now recommending your revised paper, therefore I'm pleased to inform the acceptance of your article.

·

Basic reporting

I see that the authors take into account various comments.

Experimental design

The required revisions are Well Done

Validity of the findings

The required revisions are Well Done

Additional comments

None

Reviewer 2 ·

Basic reporting

Changes previously requested by me from the authors have been made. The article is acceptable to me in this form.

Experimental design

Changes previously requested by me from the authors have been made. The article is acceptable to me in this form.

Validity of the findings

Changes previously requested by me from the authors have been made. The article is acceptable to me in this form.